# Post training 4-bit quantization of convolutional networks for rapid-deployment

Ron Banner[1] , Yury Nahshan[1] , and Daniel Soudry[2]

Intel – Artificial Intelligence Products Group (AIPG)[1]
Technion – Israel Institute of Technology[2]

{ron.banner, yury.nahshan}@intel.com
daniel.soudry@gmail.com

## Abstract

Convolutional neural networks require significant memory bandwidth and storage for intermediate computations, apart from substantial computing resources. Neural network quantization has significant benefits in reducing the amount of intermediate results, but it often requires the full datasets and time-consuming fine tuning to recover the accuracy lost after quantization. This paper introduces the first practical 4-bit post training quantization approach: it does not involve training the quantized model (fine-tuning), nor it requires the availability of the full dataset. We target the quantization of both activations and weights and suggest three complementary methods for minimizing quantization error at the tensor level, two of whom obtain a closed-form analytical solution. Combining these methods, our approach achieves accuracy that is just a few percents less the state-of-the-art baseline across a wide range of convolutional models. The source code to replicate all experiments is available on GitHub: https://github.com/submission2019/cnn-quantization.

## 1 Introduction

A significant drawback of deep learning models is their computational costs. Low precision is one of the key techniques being actively studied recently to overcome the problem. With hardware support, low precision training and inference can compute more operations per second, reduce memory bandwidth and power consumption, and allow larger networks to fit into a device.

The majority of literature on neural network quantization involves some sort of training either from scratch (Hubara et al., 2016) or as a fine-tuning step from a pre-trained floating point model (Han et al., 2015). Training is a powerful method to compensate for model's accuracy loss due to quantization. Yet, it is not always applicable in real-world scenarios since it requires the full-size dataset, which is often unavailable from reasons such as privacy, proprietary or when using an off-the-shelf pre-trained model for which data is no longer accessible. Training is also time-consuming, requiring very long periods of optimization as well as skilled manpower and computational resources.

Consequently, it is often desirable to reduce the model size by quantizing weights and activations post-training, without the need to re-train/fine-tune the model. These methods, commonly referred to as *post-training quantization*, are simple to use and allow for quantization with limited data. At 8-bit precision, they provide close to floating point accuracy in several popular models, e.g., ResNet, VGG, and AlexNet. Their importance can be seen from the recent industrial publications, focusing on quantization methods that avoid re-training (Goncharenko et al., 2018; Choukroun et al., 2019; Meller et al., 2019; Migacz, 2017).

Unfortunately, post-training quantization below 8 bits usually incurs significant accuracy degradation (Krishnamoorthi, 2018; Jacob et al., 2018). This paper focuses on CNN post-training quantization to 4-bit representation. In the absence of a training set, our methods aim at minimizing the local error introduced during the quantization process (e.g., round-off errors). To that end, we often adopt knowledge about the statistical characterization of neural network distributions, which tend to have a bell-curved distribution around the mean. This enables to design efficient quantization schemes that minimize the mean-squared quantization error at the tensor level, avoiding the need for re-training.

**Our contributions**

Our paper suggests three new contributions for post-training quantization:

1. **Analytical Clipping for Integer Quantization (ACIQ):** We suggest to limit (henceforth, clip) the range of activation values within the tensor. While this introduces distortion to the original tensor, it reduces the rounding error in the part of the distribution containing most of the information. Our method approximates the optimal clipping value analytically from the distribution of the tensor by minimizing the mean-square-error measure. This analytical threshold is simple to use during run-time and can easily be integrated with other techniques for quantization.

2. **Per-channel bit allocation:** We introduce a bit allocation policy to determine the optimal bit-width for each channel. Given a constraint on the average per-channel bit-width, our goal is to allocate for each channel the desired bit-width representation so that overall mean-square-error is minimized. We solve this problem analytically and show that by taking certain assumptions about the input distribution, the optimal quantization step size of each channel is proportional to the $\frac{2}{3}$-power of its range.

3. **Bias-correction:** We observe an inherent bias in the mean and the variance of the weight values following their quantization. We suggest a simple method to compensate for this bias.

We use ACIQ for activation quantization and bias-correction for quantizing weights. Our per-channel bit allocation method is used for quantizing both weights and activations (we explain the reasons for this configuration in Section 5). These methods are evaluated on six ImageNet models. ACIQ and bias-correction improve, on average, the 4-bit baselines by 3.2% and 6.0%, respectively. Per-channel bit allocation improves the baselines, on average, by 2.85% for activation quantization and 6.3% for weight quantization. When the three methods are used in combination to quantize both weights and activations, most of the degradation is restored without re-training, as can be seen in Figure 1.

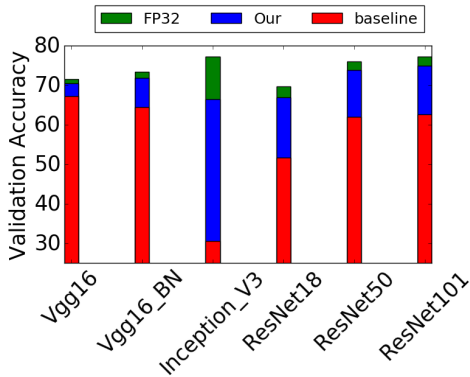

Figure 1: Top-1 accuracy of floating-point models converted directly to 4-bit weights and activations without retraining. For some models, the combination of three methods can reduce the quantization induced degradation enough as to make retraining unnecessary, enabling for the first time a rapid deployment of 4-bit models (detailed numerical results appear in Table 1).

**Previous works**

Perhaps the most relevant previous work that relates to our clipping study (ACIQ) is due to (Migacz, 2017), who also proposes to clip activations post-training. Migacz (2017) suggests a time-consuming iterative method to search for a suitable clipping threshold based on the Kullback-Leibler Divergence

(KLD) measure. This requires collecting statistics for activation values before deploying the model, either during training or by running a few calibration batches on FP32 model. It has a drawback of encountering values at runtime not obeying the previously observed statistics.

Compared to the KLD method, ACIQ avoids searching for candidate threshold values to identify the optimal clipping, which allows the clipping threshold to be adjusted dynamically at runtime. In addition, we show in the Appendix that our analytical clipping approach outperforms KLD in almost all models for 4-bit quantization even when it uses only statistical information (i.e., not tensor values observed at runtime). Zhao et al. (2019) compared ACIQ (from an earlier version of this manuscript) to KLD for higher bit-width of 5 to 8 bits. It was found that ACIQ typically outperforms KLD for weight clipping and is more or less the same for activation clipping.

Several new post-training quantization schemes have recently been suggested to handle statistical outliers. Meller et al. (2019) suggests weight factorization that arranges the network to be more tolerant of quantization by equalizing channels and removing outliers. A similar approach has recently been suggested by (Zhao et al., 2019), who suggests duplicating channels containing outliers and halving their values to move outliers toward the center of the distribution without changing network functionality. Unlike our method that focuses on 4-bit quantization, the focus of these schemes was post-training quantization for larger bitwidths.

## 2    ACIQ: Analytical Clipping for Integer Quantization

In the following, we derive a generic expression for the expected quantization noise as a function of clipping value for either Gaussian or Laplace distributions. In the Appendix, we consider the case where convolutions and rectified linear units (ReLU) are fused to avoid noise accumulation, resulting in folded-Gaussian and Laplace distributions.

Let $X$ be a high precision tensor-valued random variable, with a probability density function $f(x)$. Without loss of generality, we assume a prepossessing step has been made so that the average value in the tensor zero, i.e., $\mathbb{E}(X) = \mu = 0$ (we do not lose generality since we can always subtract and add this mean). Assuming bit-width $M$, we would like to quantize the values in the tensor uniformly to $2^M$ discrete values.

Commonly (e.g., in GEMMLOWP (Jacob et al., 2017)), integer tensors are uniformly quantized between the tensor maximal and minimal values. In the following, we show that this is suboptimal, and suggest a model where the tensor values are clipped in the range $[-\alpha, \alpha]$ to reduce quantization noise. For any $x \in \mathbb{R}$, we define the clipping function $\mathrm{clip}(x, \alpha)$ as follows

$$\mathrm{clip}(x, \alpha) = \begin{cases} x & \text{if } |x| \leq \alpha \\ \mathrm{sign}(x) \cdot \alpha & \text{if } |x| > \alpha \end{cases} \tag{1}$$

Denoting by $\alpha$ the clipping value, the range $[\alpha, -\alpha]$ is partitioned to $2^M$ equal quantization regions. Hence, the quantization step $\Delta$ between two adjacent quantized values is established as follows:

$$\Delta = \frac{2\alpha}{2^M} \tag{2}$$

Our model assumes values are rounded to the midpoint of the region (bin) i.e., for every index $i \in [0, 2^M - 1]$ all values that fall in $[-\alpha + i \cdot \Delta, -\alpha + (i+1) \cdot \Delta]$ are rounded to the midpoint $q_i = -\alpha + (2i + 1)\frac{\Delta}{2}$, as illustrated in Figure 2 left. Then, the expected mean-square-error between $X$ and its quantized version $Q(X)$ can be written as follows:

$$E[(X - Q(X))^2] =$$

$$\int_{-\infty}^{-\alpha} f(x) \cdot (x + \alpha)^2 dx + \sum_{i=0}^{2^M - 1} \int_{-\alpha + i\Delta}^{-\alpha + (i+1)\Delta} f(x) \cdot (x - q_i)^2 dx + \int_{\alpha}^{\infty} f(x) \cdot (x - \alpha)^2 dx \tag{3}$$

Eq. 3 is composed of three parts. The first and last terms quantify the contribution of $\mathrm{clip}(x, \alpha)$ to the expected mean-square-error. Note that for symmetrical distributions around zero (e.g., Gaussian $N(0, \sigma^2)$ or $\mathrm{Laplace}(0, b)$) these two terms are equal and their sum can therefore be evaluated by multiplying any of the terms by 2. The second term corresponds to the expected mean-square-error when the range $[-\alpha, \alpha]$ is quantized uniformly to $2^M$ discrete levels. This term corresponds to the

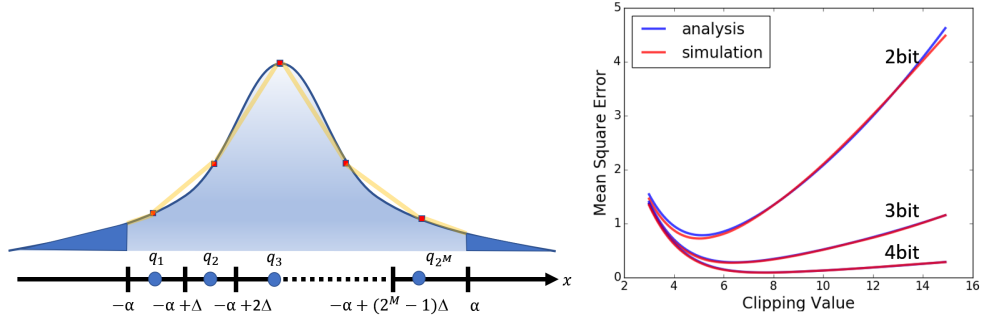

Figure 2: **left:** An activation distribution quantized uniformly in the range $[-\alpha, \alpha]$ with $2^M$ equal quantization intervals (bins) **right:** Expected mean-square-error as a function of clipping value for different quantization levels (Laplace ($\mu = 0$ and $b = 1$)). Analytical results, stated by Eq. 5, are in a good agreement with simulations, which where obtained by clipping and quantizing 10,000 values, generated from a Laplace distribution.

quantization noise introduced when high precision values in the range $[-\alpha, \alpha]$ are rounded to the nearest discrete value.

**Quantization noise:** We approximate the density function $f$ by the construction of a piece-wise linear function whose segment breakpoints are points in $f$, as illustrated on the right side of figure 2. In the appendix we use this construction to show that quantization noise satisfies the following:

$$\sum_{i=0}^{2^M-1} \int_{-\alpha+i\cdot\Delta}^{-\alpha+(i+1)\cdot\Delta} f(x) \cdot (x-q_i)^2 dx \approx \frac{2\cdot\alpha^3}{3\cdot 2^{3M}} \cdot \sum_{i=0}^{2^M-1} \frac{1}{2\alpha} = \frac{\alpha^2}{3\cdot 2^{2M}} \quad (4)$$

**Clipping noise:** In the appendix we show that clipping noise for the case of Laplace$(0, b)$ satisfies the following:

$$\int_{\alpha}^{\infty} f(x) \cdot (x-\alpha)^2 dx = \int_{-\infty}^{-\alpha} f(x) \cdot (x-\alpha)^2 dx = b^2 \cdot e^{-\frac{\alpha}{b}}$$

We can finally state Eq. 3 for the laplace case as follows.

$$E[(X - Q(X))^2] \approx 2 \cdot b^2 \cdot e^{-\frac{\alpha}{b}} + \frac{2\cdot\alpha^3}{3} \cdot \sum_{i=0}^{2^M-1} f(q_i) = 2 \cdot b^2 \cdot e^{-\frac{\alpha}{b}} + \frac{\alpha^2}{3\cdot 2^{2M}} \quad (5)$$

On the right side of figure 2, we introduce the mean-square-error as a function of clipping value for various bit widths.

Finally, to find the optimal clipping value $\alpha$ for which mean-square-error is minimized, the corresponding derivative with respect to $\alpha$ is set equal to zero as follows:

$$\frac{\partial E[(X - Q(X))^2]}{\partial \alpha} = \frac{2\alpha}{3\cdot 2^{2M}} - 2be^{-\frac{\alpha}{b}} = 0 \quad (6)$$

Solving Eq. 6 numerically for bit-widths $M = 2, 3, 4$ results with optimal clipping values of $\alpha^* = 2.83b, 3.89b, 5.03b$, respectively. In practice, ACIQ uses $\alpha^*$ to optimally clip values by estimating the Laplace parameter $b = \mathbb{E}(|X - \mathbb{E}(X)|)$ from input distribution $X$, and multiplying by the appropriate constant (e.g., 5.03 for 4 bits).

In the appendix, we provide similar analysis for the Gaussian case. We also compare the validation accuracy against the standard GEMMLOWP approach (Jacob et al., 2017) and demonstrate significant improvements in all studied models for 3-bit activation quantization.

## 3   Per-channel bit-allocation

With classical per-channel quantization, we have a dedicated scale and offset for each channel. Here we take a further step and consider the case where different channels have different numbers of bits for

precision. For example, instead of restricting all channel values to have the same 4-bit representation, we allow some of the channels to have higher bit-width while limiting other channels to have a lower bit-width. The only requirement we have is that the total number of bits written to or read from memory remains unchanged (i.e., keep the average per-channel bit-width at 4).

Given a layer with $n$ channels, we formulate the problem as an optimization problem aiming to find a solution that allocates a quota of $B$ quantization intervals (bins) to all different channels. Limiting the number of bins $B$ translates into a constraint on the number of bits that one needs to write to memory. Our goal is to minimize the overall layer quantization noise in terms of mean-square-error.

Assuming channel $i$ has values in the range $[-\alpha_i, \alpha_i]$ quantized to $M_i$ bits of precision, Eq. 5 provides the quantization noise in terms of expected mean-square-error. We employ Eq. 5 to introduce a Lagrangian with a multiplier $\lambda$ to enforce the requirement on the number of bins as follows:

$$\mathcal{L}(M_0, M_1, ..., M_n \lambda) = \sum_i \left( 2 \cdot b^2 \cdot e^{-\frac{\alpha_i}{b}} + \frac{\alpha_i^2}{3 \cdot 2^{2M_i}} \right) + \lambda \left( \sum_i 2^{M_i} - B \right) \quad (7)$$

The first term in the Lagrangian is the total layer quantization noise (i.e., the sum of mean-square-errors over all channels as defined by Eq. 5). The second term captures the quota constraint on the total number of allowed bins $B$. By setting to zero the partial derivative of the Lagrangian function $\mathcal{L}(\cdot)$ with respect to $M_i$, we obtain for each channel index $i \in [0, n-1]$ the following equation:

$$\frac{\partial \mathcal{L}(M_0, M_1, ..., M_n, \lambda)}{\partial M_i} = -\frac{2 \ln 2 \cdot \alpha_i^2}{3 \cdot 2^{2M_i}} + \lambda \cdot 2^{M_i} = 0 \quad (8)$$

By setting to zero the partial derivative of the Lagrangian function $\mathcal{L}(\cdot)$ with respect to $\lambda$ we take into account the constraint on the number of allowed bins.

$$\frac{\partial \mathcal{L}(M_0, M_1, ..., M_n, \lambda)}{\partial \lambda} = \sum_i 2^{M_i} - B = 0 \quad (9)$$

Considering Eq. 8 and Eq. 9, we have a separate equation for each channel $i \in [0, n-1]$ and an additional equation for the the Lagrangian multiplier $\lambda$. In the Appendix, we show that the solution to this system of equations results with the following simple rule for optimal bin allocation for each channel $i$:

$$B_i^\star = 2^{M_i} = \frac{\alpha_i^{\frac{2}{3}}}{\sum_i \alpha_i^{\frac{2}{3}}} \cdot B \quad (10)$$

By taking the logarithm of both sides, we translate Eq. 10 into bit width assignment $M_i$ for each channel $i$. Since $M_i$ is an integer it includes a round operation.

$$M_i = \left\lfloor \log_2 \left( \frac{\alpha_i^{\frac{2}{3}}}{\sum_i \alpha_i^{\frac{2}{3}}} \cdot B \right) \right\rceil \quad (11)$$

Figure 3 illustrates the mean-square-error in a synthetic experiment including two channels $i, j$, each having different values for $\alpha_i, \alpha_j$. Results of the experiment show that optimal allocations determined by Eq. 10 are in a good agreement with the best allocations found by the experiment. Finally, the validation accuracy of per-channel bit-allocation is compared in the appendix when activations are quantized on average to 3-bit precision. Unlike the baseline method that assigns a precision of exactly 3 bits to each channel in a layer, the per-channel bit-allocation method does not change the total bit rate to memory but significantly improve validation accuracy in all models.

## 4  Bias-Correction

We observe an inherent bias in the mean and the variance of the weight values following their quantization. Formally, denoting by $W_c \subseteq W$ the weights of channel $c$ and its quantized version by $W_c^q$, we observe that $\mathbb{E}(W_c) \neq \mathbb{E}(W_c^q)$ and $||W_c - \mathbb{E}(W_c)||_2 \neq ||W_c^q - \mathbb{E}(W_c^q)||_2$. We suggest to compensate for this quantization bias. To that end, we first evaluate correction constants for each channel $c$ as follows:

$$\mu_c = \mathbb{E}(W_c) - \mathbb{E}(W_c^q)$$
$$\xi_c = \frac{||W_c - \mathbb{E}(W_c)||_2}{||W_c^q - \mathbb{E}(W_c^q)||_2} \quad (12)$$

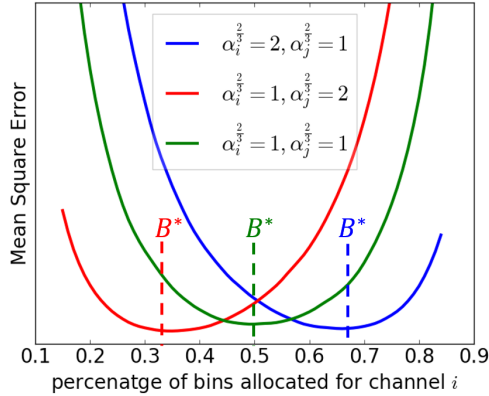

Figure 3: Optimal bin-allocation in a synthetic experiment including of a pair of channels $i, j$, each consisting of 1000 values taken from $\mathcal{N}(0, \alpha_i^2)$ and $\mathcal{N}(0, \alpha_j^2)$. The overall bin quota for the layer is set to $B = 32$, equivalent in terms of memory bandwidth to the number of bins allocated for two channels at 4-bit precision. As indicated by the vertical lines in the plot, the optimal allocations (predicted by Eq. 10) coincide with the best allocations found by the experiment.

Then, we compensate for the bias in $W_c^q$ for each channel $c$ as follows:

$$w \longleftarrow \xi_c \left( w + \mu_c \right), \quad \forall w \in W_c^q \tag{13}$$

We consider a setup where each channel has a different scale and offset (per-channel quantization). We can therefore compensate for this bias by folding for each channel $c$ the correction terms $\mu_c$ and $\xi_c$ into the scale and offset of the channel $c$. In the appendix, we demonstrate the benefit of using bias-correction for 3-bit weight quantization.

## 5 Combining our quantization methods

In the previous sections, we introduce each of the quantization methods independently of the other methods. In this section, we consider their efficient integration.

### 5.1 Applicability

We use per-channel bit allocation for both weights and activations. We found no advantage in doing any kind of weight clipping. This is in line with earlier works that also report no advantage to weight clipping for larger bitwidths (Migacz, 2017; Zhao et al., 2019). Therefore, ACIQ was considered for quantizing activations only. On the other hand, bias correction could in principle be implemented for both weights and activations. Yet, unlike bias correction for weights that can be done offline before model deployment, activation bias is estimated by running input images, which might not be available for gathering statistics at post-training . As the online alternative of estimating the activation bias on the fly during run-time might be prohibitive, we considered the bias correction method only for the weights.

### 5.2 Interaction between quantization medthods

We conduct a study to investigate how each quantization method affects performance. We consider four quantization methods: (1) ACIQ; (2) Bias-correction (3) Per-channel bit-allocation for weights; (4) Per-channel bit allocation for activations. In Figure 4, we demonstrate the interaction between these methods at different quantization levels for various models. In the appendix, we report the results of an experiment on ResNet101 where all possible interactions are evaluated (16 combinations).

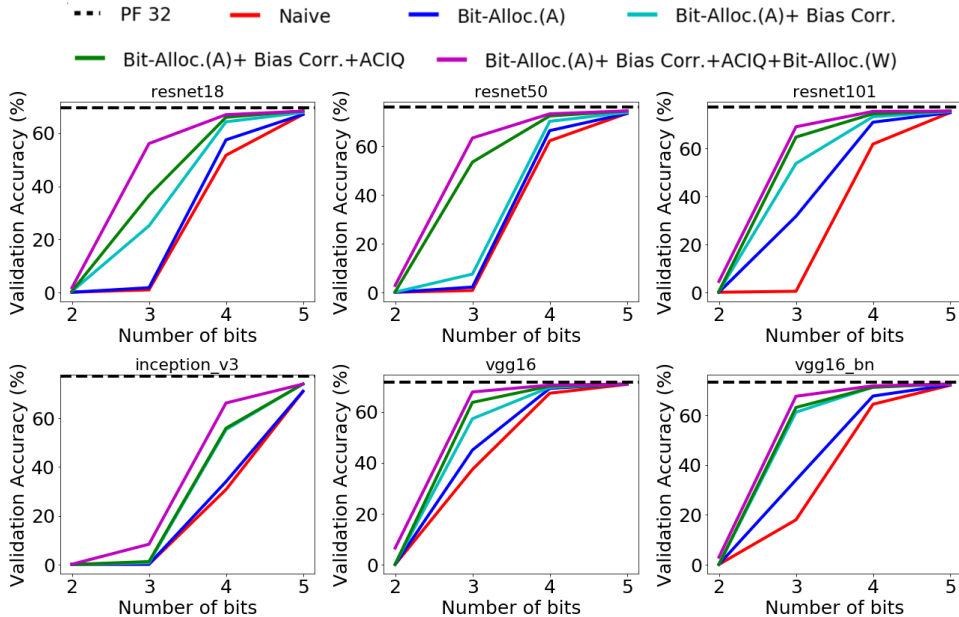

Figure 4: An ablation study showing the methods work in synergy and effectively at 3-4 bit precision.

## 6 Experiments & Results

This section reports experiments on post-training quantization using six convolutional models originally pre-trained on the ImageNet dataset. We consider the following baseline setup:

**Per-channel-quantization of weights and activations:** It is often the case where the distributions of weights and activations vary significantly between different channels. In these cases, calculating a scale-factor per channel can provide good accuracy for post-training quantization (Krishnamoorthi, 2018). The per-channel scale has shown to be important both for inference (Rastegari et al., 2016) and for training (Wu et al., 2018).

**Fused ReLU:** In convolution neural networks, most convolutions are followed by a rectified linear unit (ReLU), zeroing the negative values. There are many scenarios where these two operations can be fused to avoid the accumulation of quantization noise. In these settings, we can ignore the negative values and find an optimal clipping value $\alpha$ for the positive half space $[0, \alpha]$. Fused ReLU provides a smaller dynamic range, which leads to a smaller spacing between the different quantization levels and therefore smaller roundoff error upon quantization. In the Appendix, we provide a detailed analysis for the optimal value of $\alpha$.

We use the common practice to quantize the first and the last layer as well as average/max-pooling layers to 8-bit precision. Table 1 summarizes our results for 4-bit post training quantization. In the appendix we provide additional results for 3-bit quantization.

## 7 Conclusion

Learning quantization for numerical precision of 4-bits and below has long been shown to be effective (Lin et al., 2017; McKinstry et al., 2018; Zhou et al., 2016; Choi et al., 2018). However, these schemes pose major obstacles that hinder their practical use. For example, many DNN developers only provide the pre-trained networks in full precision without the training dataset from reasons such as privacy or massiveness of the data size. Consequently, quantization schemes involving training have largely been ignored by the industry. This gap led intensive research efforts by several tech giants and start-ups to improve post-training quantization: (1) Samsung (Lee et al., 2018), (2) Huawei, (Choukroun et al., 2019), (3) Hailo Technologies (Meller et al., 2019), (4) NVIDIA (Migacz, 2017). Our main findings in this paper suggest that with just a few percent accuracy degradation, retraining CNN models may be unnecessary for 4-bit quantization.

Table 1: ImageNet Top-1 validation accuracy with post-training quantization using the three methods suggested by this work. **Quantizing activations (8W4A):** (A) *Baseline* consists of per-channel quantization of activations and fused ReLU; each channel is quantized to 4-bit precision with a uniform quantization step between the maximum and minimum values of the channel (GEMMLOWP, Jacob et al. (2017)). (B) *ACIQ* optimally clips the values within each channel before applying quantization. (C) *Per-channel bit allocation* assigns to each activation channel an optimal bit-width without exceeding an average of 4 bits per channel, as determined by Eq. 11. (D) *ACIQ + Per channel bit allocation* quantize the activation tensors in a two stage pipeline: bit-allocation and clipping. **Quantizing weights (4W8A):** (A) *Baseline* consists of per-channel quantization of weights. (B) *Bias-correction* compensates for the quantization bias using Eq. 13. (C) *Per-channel bit allocation* assigns to each weight channel the optimal bit-width without violating the quota of allowed bits, which translates on average to 4 bits per channel (D) *Bias-Corr + Per-channel bit allocation* quantize the weight tensors in a three-stage pipeline: per-channel bit-allocation, quantization and bias correction to compensate for the quantization bias. **Quantizing weights and activation (4W4A):** Baseline consists of a combination of the above two baseline settings, i.e., (4W8A) and (8W4A). Our pipeline incorporates into the baseline all methods suggested by our work, namely, ACIQ for activation quantization, per-channel bit allocation of both weights and activations, and bias correction for weight quantization.

| Method | VGG | VGG-BN | IncepV3 | Res18 | Res50 | Res101 |
|---|---|---|---|---|---|---|
| **Quantizing activations: 8 bits weights, 4 bits activations (8W4A)** | | | | | | |
| (Per channel quantization of activations + fused ReLU) | | | | | | |
| Baseline | 68.8% | 70.6% | 70.9% | 61.5% | 68.3% | 66.5% |
| ACIQ | 70.1% | 72.0% | 72.7% | 66.6% | 71.8% | 72.6% |
| Per-channel bit allocation | 69.7% | 72.6% | 74.3% | 65.0% | 71.3% | 70.8% |
| ACIQ + Per-channel bit allocation | 70.7% | 72.8% | 75.1% | 68.0% | 73.6% | 75.6% |
| Reference (FP32) | 71.6% | 73.4% | 77.2% | 69.7% | 76.1% | 77.3% |
| **Quantizing weights: 4 bits weights, 8 bits activations (4W8A)** | | | | | | |
| (Per channel quantization of weights) | | | | | | |
| Baseline | 70.5% | 68.5% | 38.4% | 59.7% | 72.5% | 74.6% |
| Bias-Correction | 71.0% | 71.7% | 59.5% | 67.4% | 74.8% | 76.3% |
| Per-channel bit-allocation | 71.0% | 71.9% | 61.4% | 66.7% | 75.0% | 76.4% |
| Bias-Corr + Per-channel bit-allocation | 71.2% | 72.4% | 68.2% | 68.3% | 75.3% | 76.9% |
| Reference (FP32) | 71.6% | 73.4% | 77.2% | 69.7% | 76.1% | 77.3% |
| **Quantizing weights and activations: 4 bits weights, 4 bits activations (4W4A)** | | | | | | |
| (Per channel quantization of weights & activations + fused ReLU) | | | | | | |
| Baseline | 67.2% | 64.5% | 30.6% | 51.6% | 62.0% | 62.6% |
| All methods combined | 70.5% | 71.8% | 66.4% | 67.0% | 73.8% | 75.0% |
| Reference (FP32) | 71.6% | 73.4% | 77.2% | 69.7% | 76.1% | 77.3% |

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
