[Supplementary Material]

# Post training 4-bit quantization - supplementary material

## 1 ACIQ: Analytical Clipping for Integer Quantization

In the following we derive a generic expression for the expected quantization noise as a function of clipping value for either Gaussian or Laplace distributions. Let $X$ be a high precision tensor-valued random variable, with a probability density function $f(x)$. Without loss of generality, we assume a prepossessing step has been made so that the average value in the tensor zero i.e., $\mathbb{E}(X) = \mu = 0$ (we do not lose generality since we can always subtract and add this mean). Assuming bit-width $M$, we would like to quantize the values in the tensor uniformly to $2^M$ discrete values.

Commonly (e.g., in GEMMLOWP (Jacob et al., 2017)), integer tensors are uniformly quantized between the tensor maximal and minimal values. In the following we show that this is suboptimal, and suggest a model where the tensor values are clipped in the range $[-\alpha, \alpha]$ to reduce quantization noise. For any $x \in \mathbb{R}$, we define the clipping function $\text{clip}(x, \alpha)$ as follows

$$\text{clip}(x, \alpha) = \begin{cases} x & \text{if } |x| \leq \alpha \\ \text{sign}(x) \cdot \alpha & \text{if } |x| > \alpha \end{cases} \tag{A.1}$$

Denoting by $\alpha$ the clipping value, the range $[\alpha, -\alpha]$ is partitioned to $2^M$ equal quantization regions. Hence, the quantization step $\Delta$ between two adjacent quantized values is established as follows:

$$\Delta = \frac{2\alpha}{2^M} \tag{A.2}$$

Our model assumes values are rounded to the midpoint of the region (bin) i.e., for every index $i \in [0, 2^M - 1]$ all values that fall in $[-\alpha + i \cdot \Delta, -\alpha + (i+1) \cdot \Delta]$ are rounded to the midpoint $q_i = -\alpha + (2i + 1)\frac{\Delta}{2}$, as illustrated in Figure 1 left. Then, the expected mean-square-error between $X$ and its quantized version $Q(X)$ can be written as follows:

$$E[(X - Q(X))^2] =$$
$$\int_{-\infty}^{-\alpha} f(x) \cdot (x + \alpha)^2 dx + \sum_{i=0}^{2^M - 1} \int_{-\alpha + i\Delta}^{-\alpha + (i+1)\Delta} f(x) \cdot (x - q_i)^2 dx + \int_{\alpha}^{\infty} f(x) \cdot (x - \alpha)^2 dx \tag{A.3}$$

Eq. A.3 is composed of three parts. The first and last terms quantify the contribution of $\text{clip}(x, \alpha)$ to the expected mean-square-error. Note that for symmetrical distributions around zero (e.g., Gaussian $N(0, \sigma^2)$ or $\text{Laplace}(0, b)$) these two terms are equal and their sum can therefore be evaluated by multiplying any of the terms by 2. The second term corresponds to the expected mean-square-error when the range $[-\alpha, \alpha]$ is quantized uniformly to $2^M$ discrete levels. This terms corresponds to the quantization noise introduced when high precision values in the range $[-\alpha, \alpha]$ are rounded to the nearest discrete value.

## 1.1 Quantization noise

We approximate the density function $f$ by a construction of a piece-wise linear function whose segment breakpoints are points in $f$, as illustrated on the right side of figure 1. Since we consider only smooth probability density functions (e.g., Gaussian or Laplace), the resulting approximation error is small for sufficient resolution i.e., small quantization step size $\Delta$. In section 2 in this appendix we show that given a density function $f$, the quantization noise can be approximated as follows:

$$\sum_{i=0}^{2^M-1} \int_{-\alpha+i\cdot\Delta}^{-\alpha+(i+1)\cdot\Delta} f(x)\cdot(x-q_i)^2 dx \approx \frac{2\cdot\alpha^3}{3\cdot2^{3M}}\cdot\sum_{i=0}^{2^M-1} f(q_i) \tag{A.4}$$

Eq. A.4 represents the rounding error (as opposed to clipping error) due to the rounding of all values in the bin $i$ to its center $q_i$. For sufficient resolution and a smooth density function, the density function $f$ can be approximated by a uniform distribution in the range $[-\alpha, \alpha]$ (Marco & Neuhoff, 2005), which enables much simpler analysis with little effect on the accuracy. In Figure 1, we show that with this assumption the analytic results are in a good agreement with the simulation results. By substituting the uniform density function $f(x) = \frac{1}{2\alpha}$ into eq. A.4, the following simpler rounding error can be computed:

$$\sum_{i=0}^{2^M-1} \int_{-\alpha+i\cdot\Delta}^{-\alpha+(i+1)\cdot\Delta} f(x)\cdot(x-q_i)^2 dx \approx \frac{2\cdot\alpha^3}{3\cdot2^{3M}}\cdot\sum_{i=0}^{2^M-1}\frac{1}{2\alpha} = \frac{\alpha^2}{3\cdot2^{2M}} \tag{A.5}$$

By substituting eq. A.5 into eq. A.3, and using the symmetrical argument mentioned above, eq. A.3 can be simplified for symmetrical distributions as follows:

$$E[(X-Q(X))^2] = \frac{\alpha^2}{3\cdot2^{2M}} + 2\cdot\int_{\alpha}^{\infty} f(x)\cdot(x-\alpha)^2 dx \tag{A.6}$$

In the following we provide a closed form solution for the case where the density probability distribution function $f(x)$ is either Gaussian $N(0,\sigma^2)$ or $\text{Laplace}(0,b)$.

## 1.2 Clipping noise

In the following we develop an expression based on eq. A.6 for the Laplace case. In Section 3 in this appendix we provide a similar analysis for the case where the probability density function is Gaussian $N(0,\sigma^2)$

Assuming $\mu = 0$, we have the following Laplace density function $f(x) = \frac{1}{2b}\mathrm{e}^{-\frac{|x|}{b}}$. In order to derive a closed form solution for eq. A.6, we need to evaluate

$$2\cdot\int_{\alpha}^{\infty} f(x)\cdot(x-\alpha)^2 dx. \tag{A.7}$$

Let $\Psi(x)$ represent the expression below:

$$\Psi(x) = \frac{\mathrm{e}^{-\frac{x}{b}}}{2}\left[2\alpha b - 2b^2 - \alpha^2 - x^2 - 2(b-\alpha)x\right] \tag{A.8}$$

By taking the derivative of $\Psi(x)$ with respect to $x$, it is easy to see that $\Psi(x)$ is the correct antiderivative of the integrand in eq. A.7. Hence,

$$\int_{\alpha}^{\infty} f(x)\cdot(x-\alpha)^2 dx = \Psi(\inf) - \Psi(\alpha) = b^2\cdot e^{-\frac{\alpha}{b}}$$

We can finally state eq. A.6 for the laplace case as follows.

$$E[(X-Q(X))^2] \approx 2\cdot b^2\cdot e^{-\frac{\alpha}{b}} + \frac{2\cdot\alpha^3}{3}\cdot\sum_{i=0}^{2^M-1} f(q_i) = 2\cdot b^2\cdot e^{-\frac{\alpha}{b}} + \frac{\alpha^2}{3\cdot2^{2M}} \tag{A.9}$$

Figure 1: **left:** Expected mean-square-error as a function of clipping value for different quantization levels (Laplace ($\mu = 0$ and $b = 1$)). Analytical results, stated by eq. A.9, are in a good agreement with simulations, which where obtained by clipping and quantizing 10,000 values, generated from a Laplace distribution. **right:** An activation distribution quantized uniformly in the range $[-\alpha, \alpha]$ with $2^M$ equal quantization intervals (bins)

On the left side of figure 1, we introduce the mean-square-error as a function of clipping value for various bit widths. Finally, to find the optimal clipping value $\alpha$ for which mean-square-error is minimized, the corresponding derivative with respect to $\alpha$ is set equal to zero as follows:

$$\frac{\partial E[(X - Q(X))^2]}{\partial \alpha} = \frac{2\alpha}{3 \cdot 2^{2M}} - 2be^{-\frac{\alpha}{b}} = 0 \tag{A.10}$$

Solving Eq. A.10 numerically for bit-widths $M = 2, 3, 4$ results with optimal clipping values of $\alpha^* = 2.83b, 3.89b, 5.03b$, respectively.

## 2  ACIQ: Piece-Wise Linear Approximation

Here we provide a more accurate analysis related to the qunatization noise (i.e., the second term in Equation A.3), measured as the expected mean-square-error when the range $[-\alpha, \alpha]$ is quantized uniformly to $2^M$ discrete levels. To that end, we approximate the density function $f$ by a construction of a piece-wise linear function $g$ such that $f(q_i) = g(q_i)$ for each $i \in [0, 2^M - 1]$. Since we consider only smooth probability density functions (e.g., Gaussian or Laplace), the resulting approximation error is small for sufficient resolution i.e., small quantization step size $\Delta$. In figure 1 we provide an illustration for this construction.

We turn to calculate the linear equation for each line segment of the piece-wise linear function $g$, falling in the range $[-\alpha + i \cdot \Delta, -\alpha + (i + 1) \cdot \Delta]$. To that end, we consider the slope (derivative) and the value of the density function at the midpoint $q_i$. With these two values we can define for each segment $i \in [0, 2^M - 1]$ the corresponding form of linear approximation:

$$g(x) = f(q_i) + \frac{df}{dx}(q_i) \cdot (x - q_i), \text{where } x \in [-\alpha + i \cdot \Delta, -\alpha + (i + 1) \cdot \Delta] \tag{A.11}$$

We now turn to calculate the second term in Equation A.3. By equation A.11, and since $q_i$ is defined to be the midpoint between the integration limits, the following holds true

$$\sum_{i=0}^{2^M-1} \int_{-\alpha+i\cdot\Delta}^{-\alpha+(i+1)\cdot\Delta} f(x) \cdot (x - q_i)^2 dx \approx \sum_{i=0}^{2^M-1} \int_{-\alpha+i\cdot\Delta}^{-\alpha+(i+1)\cdot\Delta} g(x) \cdot (x - q_i)^2 dx =$$

$$= \sum_{i=0}^{2^M-1} \int_{-\alpha+i\cdot\Delta}^{-\alpha+(i+1)\cdot\Delta} f(q_i) \cdot (x - q_i)^2 + \sum_{i=0}^{2^M-1} \int_{-\alpha+i\cdot\Delta}^{-\alpha+(i+1)\cdot\Delta} \frac{df}{dx}(q_i) \cdot (x - q_i)^3 dx =$$

$$= \frac{1}{3} \sum_{i=0}^{2^M-1} f(q_i) \cdot (x - q_i)^3 \Big|_{-\alpha+i\cdot\Delta}^{-\alpha+(i+1)\cdot\Delta} + \frac{1}{4} \sum_{i=0}^{2^M-1} \frac{df}{dx}(q_i) \cdot (x - q_i)^4 \Big|_{-\alpha+i\cdot\Delta}^{-\alpha+(i+1)\cdot\Delta} =$$

$$= \frac{\Delta^3}{12} \sum_{i=0}^{2^M-1} f(q_i) = \frac{2 \cdot \alpha^3}{3 \cdot 2^{3M}} \cdot \sum_{i=0}^{2^M-1} f(q_i)$$

## 3 Clipping noise (Gaussian case)

We now turn to evaluate Equation A.6 for the Gaussian case. Given a Gaussian random variable $X \sim N(0, \sigma^2)$, we define $\Psi(x)$ to represent the expression below:

$$\Psi(x) = \frac{\left(\alpha^2 + \sigma^2\right) \operatorname{erf}\left(\frac{x}{\sqrt{2}\sigma}\right)}{2} - \frac{(x\sigma - 2\alpha\sigma)\,\mathrm{e}^{-\frac{x^2}{2\sigma^2}}}{\sqrt{2\pi}} \tag{A.12}$$

As in subsection 1.2, one can observe that by taking the derivative of $\Psi(x)$ with respect to $x$, it is easy to show that $\Psi(x)$ is the correct antiderivative of Equation A.7 for the case where $f$ represents the Gaussian density function i.e., $f(x) = \frac{1}{\sqrt{2\pi}\sigma}\mathrm{e}^{-\frac{x^2}{2\sigma^2}}$. Next, we use $\Psi(x)$ on the range $[\alpha, \infty]$ to evaluate Equation A.7 for the Gaussian case as follows:

$$\int_\alpha^\infty f(x) \cdot (x - \alpha)^2 dx = \Psi(\infty) - \Psi(\alpha) = \frac{\alpha^2 + \sigma^2}{2} \cdot \left[1 - \operatorname{erf}\left(\frac{\alpha}{\sqrt{2}\sigma}\right)\right] - \frac{\alpha \cdot \sigma \cdot \mathrm{e}^{-\frac{\alpha^2}{2 \cdot \sigma^2}}}{\sqrt{2\pi}}$$

Equation A.6 can thus be written for the case of Gaussian distribution as follows:

$$E[(X - Q(X))^2] \approx\approx (\alpha^2 + \sigma^2) \cdot \left[1 - \operatorname{erf}\left(\frac{\alpha}{\sqrt{2}\sigma}\right)\right] + \frac{\alpha^2}{3 \cdot 2^{2M}} - \frac{\sqrt{2}\alpha \cdot \sigma \cdot \mathrm{e}^{-\frac{\alpha^2}{2 \cdot \sigma^2}}}{\sqrt{\pi}} \tag{A.13}$$

In figure 2 we introduce the mean-square-error as a function of clipping value for various bit widths.

Figure 2: Expected mean-square-error as a function of clipping value for different quantization levels (Gaussian ($\mu = 0$ and $\sigma = 1$)). Analytical results , stated by Equation A.19, are in a good agreement with simulations, which where obtained by clipping and quantizing 10,000 values, generated from a Laplace distribution. As expected, the difference occurs only for very low-bit width and large clipping values where the uniform assumption tends to break.

In order to find the optimal clipping values for which mean-square-error is minimized, we need to differentiate $E[(X - Q(X))^2]$ with respect to $\alpha$ and set the derivative equal to zero as follows.

$$\frac{\partial E[(X - Q(X))^2]}{\partial \alpha} = \alpha \left[1 - \operatorname{erf}\left(\frac{\alpha}{\sqrt{2}\sigma}\right)\right] - \frac{\sigma^2 \mathrm{e}^{-\frac{\alpha^2}{2\sigma^2}}}{\sqrt{2\pi}\sigma} - \frac{\sigma \mathrm{e}^{-\frac{\alpha^2}{2\sigma^2}}}{\sqrt{2\pi}} + \frac{2\alpha}{3 \cdot 2^{2M}} = 0 \tag{A.14}$$

## 4 ACIQ: Optimal Quantizer for Fused ReLU Activations

In this section we adjust Equations A.9 and A.13 for the case where convolutions and rectified linear units (ReLU) are fused to avoid accumulation of noise.

The ReLU is defined by zeroing the negative half space i.e., $g(x) = \max(0, x)$. Given a high precision random variable $X$ with a probability density function $f(x)$ we would like to minimize the following expected mean square-error

$$E\left[\left(g(X) - Q(g(X))\right)^2\right] \tag{A.15}$$

Assuming the probability density function $f(x)$ has a symmetrical distribution around zero, there are two adjustments that need to be made in the analysis of Section 1:

(1) The quantization step $\Delta$ is now set according to the range $[0, \alpha]$. Hence, Equation A.2 should be modified as follows:

$$\Delta = \frac{\alpha}{2^M} \tag{A.16}$$

(2) Since we consider only the positive values, Equation A.3 should ignore the negative contribution i.e.,

$$E[(X - Q(X))^2] = \sum_{i=0}^{2^M - 1} \int_{i \cdot \Delta}^{(i+1) \cdot \Delta} f(x) \cdot (x - q_i)^2 dx + \int_{\alpha}^{\infty} f(x) \cdot (x - \alpha)^2 dx \tag{A.17}$$

This translates to the following adjustments in Equation A.9 for the Laplace case:

$$E\left[\left(g(X) - Q(g(X))\right)^2\right] \approx b^2 \cdot e^{\frac{-\alpha}{b}} + \frac{\alpha^2}{24 \cdot 2^{2M}} \tag{A.18}$$

Similarly, for the Gaussian case Equation A.13 is modified as follows:

$$E\left[\left(g(X) - Q(g(X))\right)^2\right] \approx \frac{\alpha^2 + \sigma^2}{2} \cdot \left[1 - \text{erf}\left(\frac{\alpha}{\sqrt{2}\sigma}\right)\right] + \frac{\alpha^2}{24 \cdot 2^{2M}} - \frac{\alpha \cdot \sigma \cdot e^{-\frac{\alpha^2}{2 \cdot \sigma^2}}}{\sqrt{2\pi}} \tag{A.19}$$

# 5 Per-channel bit-allocation

With classical per-channel quantization we have a dedicated scale and an offset for each channel. Here we take a further step and consider the case where different channels have different numbers of bits for precision. For example, instead of restricting all channel values to have the same 4-bit representation, we allow some of the channels to have higher bit-width while limiting other channels to have a lower bit-width. So the total volume of data written to or read memory is still comparable to 4-bit precision.

Given a layer with $n$ channels, we formulate the problem as an optimization problem aiming to find a solution that allocates a quota of $B$ quantization intervals (bins) to all different channels. Limiting the number of bins $B$ translates into a constraint on the number of bits that one needs to write to memory. Our goal is to minimize the overall layer quantization noise in terms of mean-square-error.

Assuming channel $i$ has values in the range $[-\alpha_i, \alpha_i]$ quantized to $M_i$ bits of precision, eq. A.9 provides the quantization noise in terms of expected mean-square-error. We employ eq. A.9 to introduce a Lagrangian with a multiplier $\lambda$ to enforce the requirement on the number of bins as follows:

$$\mathcal{L}(M_0, M_1, ..., M_n \lambda) = \sum_i \left(2 \cdot b^2 \cdot e^{-\frac{\alpha_i}{b}} + \frac{\alpha_i^2}{3 \cdot 2^{2M_i}}\right) + \lambda \left(\sum_i 2^{M_i} - B\right) \tag{A.20}$$

The first term in the Lagrangian is the total layer quantization noise (i.e., the sum of mean-square-errors over all channels as defined by eq. A.9). The second term captures the quota constraint on the total number of allowed bins $B$. By setting to zero the partial derivative of the Lagrangian function $\mathcal{L}(\cdot)$ with respect to $M_i$, we obtain for each channel index $i \in [0, n-1]$ the following equation:

$$\frac{\partial \mathcal{L}(M_0, M_1, ..., M_n \lambda)}{\partial M_i} = -\frac{2 \ln 2 \cdot \alpha_i^2}{3 \cdot 2^{2M_i}} + \lambda \cdot 2^{M_i} = 0 \tag{A.21}$$

Hence,

$$2^{M_i} = \sqrt[3]{\frac{2 \ln 2 \cdot \alpha_i^2}{3 \cdot \lambda}} \tag{A.22}$$

Next, by setting to zero the partial derivative of the Lagrangian function $\mathcal{L}(\cdot)$ with respect to $\lambda$ we take into account the constraint on the number of allowed bins.

$$\frac{\partial \mathcal{L}(M_0, M_1, ..., M_n\lambda)}{\partial \lambda} = \sum_i 2^{M_i} - B = 0 \tag{A.23}$$

Hence, using eq. A.22 we get the following expression:

$$\sum_i 2^{M_i} = \sum_i \sqrt[3]{\frac{2\ln 2 \cdot \alpha_i^2}{3 \cdot \lambda}} = \sqrt[3]{\frac{2\ln 2}{3\lambda}} \cdot \sum_i \alpha_i^{\frac{2}{3}} = B \tag{A.24}$$

Hence,

$$\lambda = \frac{2\ln 2}{3B^3} \cdot \left( \sum_i \alpha_i^{\frac{2}{3}} \right)^3 \tag{A.25}$$

Define $B^\star$ to be an optimal bin allocation that minimizes the mean-square-error. By substituting eq. A.25 into eq. A.22, we get the following simple rule for optimal bin allocation for each channel $i$:

$$B_i^\star = 2^{M_i} = \frac{\alpha_i^{\frac{2}{3}}}{\sum_i \alpha_i^{\frac{2}{3}}} \cdot B \tag{A.26}$$

Finally, by taking the logarithm of both sides, we translate eq. A.26 into bit width assignment $M_i$ for each channel $i$. Since $M_i$ is an integer it includes a round operation.

$$M_i = \left\lfloor \log_2 \left( \frac{\alpha_i^{\frac{2}{3}}}{\sum_i \alpha_i^{\frac{2}{3}}} \cdot B \right) \right\rceil \tag{A.27}$$

## 6 Kullback Leibler divergence (KLD) method

The following table summaries the classification test accuracies of different popular pre-trained convolution networks after activations are quantized to 4-bit precision in a post-training manner (8W4A). Due to scaling issues of KLD, we could not test its performance in conjecture with the other search-based quantizaton schemes (e.g., KLD with per-channel quantization). Therefore,to make a fair comparison we compare against a baseline that does not include per-channel quantization.

| Model | Naive (8W4A) | KLD (8W4A) | ACIQ (8W4A) | Reference (float32) |
|---|---|---|---|---|
| VGG16 | 53.90% | 67.04% | 67.40% | 71.59% |
| VGG16-BN | 29.50% | 65.85% | 67.60% | 73.36% |
| ResNet-18 | 53.20% | 65.06% | 65.80% | 69.75% |
| ResNet-50 | 52.70% | 70.80% | 71.45% | 76.10% |
| ResNet-101 | 50.80% | 71.70% | 69.53% | 77.30% |
| Inception v3 | 41.40% | 59.25% | 60.80% | 77.20% |
| AlexNet | 41.60% | 49.55% | 52.20% | 56.52% |

Table 1: Validation accuracy of various architectures quantized post-training to 8-bit weights and 4-bit activations (8W4A): *Naive (8W4A)* refers to the conventional quantization method based on the maximum and minimum representable value which shows severe accuracy loss. *KLD (8W4A)* refers to the iterative method suggested by NVIDIA to search for a good clipping threshold based on the Kullback-Leibler Divergence measure Migacz (2017). *ACIQ (8W4A)* refers to our analytic clipping approach described in Section 1; unlike KLD, which is a brute force technique, our approach is order of times faster, and, excluding ResNet 101, maintains higher validation accuracy. *Reference (float32)* uses full precision models with 32 bit weights and activations.

## 7 Results for 3-bit Quantization

We compared ACIQ, per-channel bit allocation and bias correction also for the case of 3-bit precision. Our results are summarized in Table 2.

Table 2: ImageNet Top-1 validation accuracy with post-training quantization using the three methods suggested by this work.

| Method | VGG | VGG-BN | IncepV3 | Res18 | Res50 | Res101 |
|---|---|---|---|---|---|---|
| **Quantizing activations: 8 bits weights, 3 bits activations (8W3A)** | | | | | | |
| (Per channel quantization of activations + fused ReLU) | | | | | | |
| Baseline | 57.1% | 56.0% | 34.1% | 23.4% | 5.6% | 1.6% |
| ACIQ | 67.0% | 69.1% | 56.8% | 57.8% | 60.8% | 59.0% |
| Per-channel bit allocation | 64.0% | 69.7% | 55.2% | 48.7% | 16.2% | 61.5% |
| Reference (FP32) | 71.6% | 73.4% | 77.2% | 69.7% | 76.1% | 77.3% |
| **Quantizing weights: 3 bits weights, 8 bits activations (3W8A)** | | | | | | |
| (Per channel quantization of weights) | | | | | | |
| Baseline | 59.6% | 40.4% | 0% | 3.8% | 28.6% | 50.5% |
| Bias-Correction | 67.3% | 66.1% | 3.0% | 43.5% | 67.4% | 70.7% |
| Per-channel bit-allocation | 69.5% | 63.6% | 1.3% | 44.0% | 66.6% | 72.6% |
| Reference (FP32) | 71.6% | 73.4% | 77.2% | 69.7% | 76.1% | 77.3% |

# 8 Combining our Quantization Methods

We conduct a study to investigate how each quantization method affects performance. We consider four quantization methods: (1) ACIQ; (2) Bias-correction (3) Per-channel bit-allocation for weights; (4) Per-channel bit allocation for activations. Table 3 summaries all possible combinations on ResNet101.

| Method | 2 bit | 3 bit | 4 bit | 5 bit |
|---|---|---|---|---|
| Naive | 0.1 | 0.5 | 61.8 | 74.9 |
| Bias-Corr. | 0.2 | 0.9 | 63.7 | 75.3 |
| Bit-Alloc.(W) | 0.1 | 1.3 | 63 | 74.9 |
| Bit-Alloc.(W) + Bias Corr | 0.1 | 1.3 | 65.5 | 75.3 |
| Bit-Alloc.(A) | 0.1 | 31.7 | 70.9 | 74.9 |
| Bit-Alloc.(A) + Bias Corr | 0.4 | 53.7 | 73.2 | 75.3 |
| Bit-Alloc.(A) + Bit-Alloc.(W) | 0.2 | 56.2 | 70.9 | 74.9 |
| Bit-Alloc.(A) + Bit-Alloc.(W) + Bias-Corr. | 0.6 | 58 | 72.4 | 75.4 |
| ACIQ | 0 | 23.3 | 68.2 | 75 |
| Bias-Corr. + ACIQ | 0.3 | 47.7 | 71 | 75.6 |
| Bit-Alloc.(W) + ACIQ | 0.2 | 53.7 | 71.1 | 75 |
| Bit-Alloc.(W) + Bias Corr + ACIQ | 0.5 | 55.4 | 72.2 | 75.5 |
| Bit-Alloc.(A) + ACIQ | 0 | 41.7 | 72.1 | 75.2 |
| Bit-Alloc.(A) + Bias Corr + ACIQ | 0.5 | 64.7 | 74.4 | 75.6 |
| Bit-Alloc.(A) + Bit-Alloc.(W) + ACIQ | 0.4 | 66.6 | 74.8 | 75.2 |
| Bit-Alloc.(A) + Bit-Alloc.(W) + Bias-Corr. + ACIQ | 4.6 | 69 | 75.4 | 75.5 |

Table 3: ImageNet Top-1 validation accuracy with post-training quantization using the three methods suggested by this work.