[Reviews · NeurIPS 2019]

Reviewer 1



The authors' provide an approach for post-training quantization deep convolutional neural networks down to 4-bit weights and activations. Their approach includes several methods: 1. Analytical Clipping for Integer Quantization (ACIQ): ACIQ clips activations to minimize MSE, assuming Gaussian or Laplace distributions. 2. Per-channel bit allocation (PCBA): In this scheme one channel that needs a greater number of bits can take them from a channel that requires fewer as long as the average number of bits remains at the target (e.g., 4 bits). 3. Bias correction: This method analyse and compensates for some of the weight quantization induced bias. The authors created many combinations these approaches, applying them to ImageNet on many networks at a few activation and weight bit precisions. The authors are among the first to apply post-training quantization approaches to 4-bit and lower precision networks. The paper is interesting, and it has clear significance, creating a state of the art for post-training quantization at 4 bits. Table 1 is clear and shows how different methods effect results. The writing and results are mostly clear, but could be improved. The work may overstate its value a bit. The assumption that networks will not be retrained seems week. If there is great value in a 4-bit network, then it will be fine-tuned to achieve the best score it can. The per-channel bit allocation is difficult for inference. The benefit is reduced model size when applied to weights. Hardware must support the worst-case bit widths, so there is no real benefit for activations, so 4W4A is somewhat misleading. Also, model parsing is more complex. line 187: "5.2 Interaction between quantization medthods" should be "methods"

Reviewer 2



In this paper, authors introduce simple yet effective dataset-free heuristics for post-training quantization. Namely, 1. Clipping the activation values in some range which helps to focus on more dense area of values and better quantize them (with less distortion). The range values are determined using mean-square-error between the original weights and quantized weights. 2. Per-channel bit-allocation. Authors propose dynamic number of bits allocation instead of fixing it ahead of time for all channels. This is done by formulating an optimization problem and solution can be obtained analytically. 3. Since quantized weights have different mean and std than the original float32 weights, authors propose to correct those differences. It can be clearly seen that these heuristics based on statistical information about weights/activations and they can be combined together. The paper is well orginized and easy to follow. All proofs and derivations are seem correct to me. Major concerns: - For ACIQ, as authors stated the idea of using clipping is not a new. It is not clear what is the performance compare to other types of removal of outliers (e.g. simply removing all values which are greater than \pm 2\sigma). - Authors apply the proposed methods for channel-wise quantization. What is the performance for other types of quantization (filter-wise, layer-wise)? Probably, some additional experiments and comparisons required to see if the methods are generally applicable. Otherwise, it might be problematic to integrate with other techniques for quantization. - What is the motivation for bias-correction? Empirically, it shows the benefits but it is unclear why having bias in the mean and variance is harmful for quantization and why this correction should improve it? ----------------------------------------------------------------------------------------------------------- Since authors address most of my concerns, I would like to increase my score from 6->7. Overall, I think that it is a good paper and has a significant contribution to the machine learning community.

Reviewer 3



1) The paper is clean, focused and novel. Directly applicable to various area and research, well reflecting the current trend on quantized neural networks. 2) Can you explain why ACIQ on InceptionV3 is less effective than Resnet-50 or Resnet-101? Is it related to distribution assumption? 3) For clarity, it would better specify "signed" or "unsigned" quantization. When using activation quantization after ReLU, using [0, a] range and "unsigned" 8/4-bit for quantization. In this case, "round to midpoint" also valid?

[Author Response · NeurIPS 2019]

We thank all the reviewers for their helpful feedback and remarks, and for being unanimously positive about the
significance of the results: R1— *"The authors are among the first to apply post-training quantization approaches to*
*4-bit and lower precision networks. The paper is interesting, and it has clear significance, creating a state of the art*
*for post-training quantization at 4 bits."*; R2— *"These contributions have a high level of significance since with small*
*overhead the method improves over standard quantization scheme."*; R3— *"The paper is clean, focused and novel.*
*Directly applicable to various area and research, well reflecting the current trend on quantized neural networks."*

**New results:** Since submission, we combined our three-stage quantization pipeline with a loss-less compression scheme
that uses variable-length codewords to encode the quantized values (Huffman encoding). Thus, more common values
are assigned with fewer bits, which has two important advantages: (i) channels with different numerical precisions
can be combined in the same layer, enabling reduced hardware overhead for per-channel bit allocation; (ii) significant
reductions in memory bandwidth requirements could be achieved, with a minor accuracy loss (at most 0.5% of the
float32 baseline). For example, the average number of bits required to represent an activation value in a feature map
could be reduced to 2.2, 2.4, 3.1, 3.7, 4.1 and 4.4 bits for VGG, VGG-BN, Inception, Res18, Res50 and Res101,
respectively. We note for the sake of clarity that these new results were obtained in a post-training manner. We have
submitted the new code to the git repository and plan to update the final version of the paper.

Below we address the main suggestions for improvements mentioned by the reviewers (minor issues will be addressed).
If we address the comments of the reviewers, we kindly ask that they adjust their scores to reflect their positive opinion.

**Referee 1:** *"My interpretation is that everything in this paper is post-training and pre-inference ...If not, it would*
*improve the clarity to state what is computed during runtime."* $\implies$ The optimal clipping threshold and channel bit-width
allocations are calculated at runtime using Equations 6 and 11, respectively. We implicitly mention that in lines 72-73
but will add a more explicit note. Our recent simulations show that by running 32 calibration images to gather activation
statistics, the above two calculations can be done offline with little effect on overall accuracy. For 4-bit weights and
activations (4W4A), overall accuracy is still significantly improved over 4W4A baseline in all models: 61.8% vs 69.8%
(VGG); 59.8% vs. 71% (VGG-BN); 22.8% vs. 66.1% (Inception V3); 46.1% vs. 65.1% (Res18); 62.2% vs. 74.2%
(Res50); 64.3% vs. 76.1% (Res101). Git repository was updated with this important use-case.

*"There are many fundamental questions outside of the current scope of this paper... 1. What is limit to quantization*
*without training? 2. Are we close or can they be much better with better methods?"* $\implies$ These are exciting open
questions. Our recent results (mentioned above) indicate we can get very low for some models e.g., 2.2 bits for VGG16.

**Referee 2:** *"It is not clear what is the performance compare to other types of removal of outliers (e.g. simply removing*
*all values which are greater than $\pm 2\sigma$)."* $\implies$ we compare and state the advantages of ACIQ over (Migacz, 2017) and
(Zhao et al., 2019) in lines 72-78 and provide a comparison against KLD in the Appendix (Table 1). Other clipping
thresholds hurts accuracy, e.g. clipping values outside the interval $[-2\sigma, 2\sigma]$ is associated with a significantly inferior
outcome compared to ACIQ in all models: 47.2% vs 70.1% (VGG); 44.9% vs. 72.1% (VGG-BN); 13.4% vs. 72.5%
(Inception V3); 25.6% vs. 66.6% (Res18); 15.4% vs. 72% (Res50); 24.7% vs. 72.7% (Res101). Git repository was
updated with this test.

*"What is the performance for other types of quantization (filter-wise, layer-wise)?"* $\implies$ Applying ACIQ for layer-wise
quantization improves results in all models: 64.3% vs. 70% (Res101); 67% vs. 71.2% (Res50); 63.7% vs. 66.5%
(Res18); 70.8% vs. 72.8% (Inception V3); 69.1% vs. 71.5% (VGG-BN); 69.9% vs 70.6% (VGG). Git repository was
updated with this test, and we plan to include this layer-wise comparison in the final version.

*"What is the motivation for bias-correction?"* $\implies$ Neural networks are known to be sensitive to shift in mean and
variance which builds up across layers, shifting all network statistics away from the learned distribution. If quantized
weights are unbiased with respect to original weights, the quantization error is unbiased and possesses the desirable
property that the *expected* rounding error is zero and thus cancels out in the layer's output.

**Referee 3:** *"during convolution, accumulation of different channel should be first matching step size, then accumulate.*
*How do you match step size, and where do you accumulate? (Int32? Float32?)"* $\implies$ per-channel quantization is
handled as follows: (1) load activations and weights from memory at 4-bit precision; (2) multiply the 4-bit weights with
4-bit activations, which results with 8-bit products; (3) expand these 8-bit products into 16-bit representations (needed
for matching the different channel quantization step sizes); (4) finally, sum the products with Int32 accumulator.

*"Is retraining after ACIQ better than without ACIQ? If yes, ACIQ may be the standard procedure for low-bit quantiza-*
*tion."* $\implies$ This is a good question. We are planning to investigate the applicability of our techniques in full training
mode and will include this question in our study.

*A difference visualization of the quantization step of the same tensor, each found by KLD and ACIQ.* $\implies$ Will add.
Compared to KLD, ACIQ has on average 15-20% smaller quantization step. Defining the mean quantization step ratio
$\rho = \frac{\Delta_{ACIQ}}{\Delta_{KLD}}$ across all layers, we get $\rho = 0.81, 0.81, 0.8, 0.86$ for Res18, Res50, Res101 and Inception, respectively.

[Meta-Review · NeurIPS 2019]

All reviewers were positive about the contributions in the paper so I recommend acceptance. Congratulations! Please take into account all the reviewers' comments when preparing the final version of the paper.